# Multiple Compounds Recognition from The Tandem Mass Spectral Data Using Convolutional Neural Network

**DOI:** 10.3390/molecules24244590

**Published:** 2019-12-15

**Authors:** Jiali Lv, Jian Wei, Zhenyu Wang, Jin Cao

**Affiliations:** 1School of Software and Microelectronics, Peking University, 24 Jinyuan Road, Daxing District, Beijing 102600, China; drake@pku.edu.cn (J.L.); 1801210439@pku.edu.cn (J.W.); 2National institutes for food and drug control, Beijing 100050, China; caojin@nifdc.org.cn

**Keywords:** tandem mass spectra, compounds recognition, multi-label classification, convolutional neural network

## Abstract

Mixtures analysis can provide more information than individual components. It is important to detect the different compounds in the real complex samples. However, mixtures are often disturbed by impurities and noise to influence the accuracy. Purification and denoising will cost a lot of algorithm time. In this paper, we propose a model based on convolutional neural network (CNN) which can analyze the chemical peak information in the tandem mass spectrometry (MS/MS) data. Compared with traditional analyzing methods, CNN can reduce steps in data preprocessing. This model can extract features of different compounds and classify multi-label mass spectral data. When dealing with MS data of mixtures based on the Human Metabolome Database (HMDB), the accuracy can reach at 98%. In 600 MS test data, 451 MS data were fully detected (true positive), 142 MS data were partially found (false positive), and 7 MS data were falsely predicted (true negative). In comparison, the number of true positive test data for support vector machine (SVM) with principal component analysis (PCA), deep neural network (DNN), long short-term memory (LSTM), and XGBoost respectively are 282, 293, 270, and 402; the number of false positive test data for four models are 318, 284, 198, and 168; the number of true negative test data for four models are 0, 23, 7, 132, and 30. Compared with the model proposed in other literature, the accuracy and model performance of CNN improved considerably by separating the different compounds independent MS/MS data through three-channel architecture input. By inputting MS data from different instruments, adding more offset MS data will make CNN models have stronger universality in the future.

## 1. Introduction

Mass spectrometry is one of the most powerful tools for pure substances’ identifications [1]. The mass spectral data contains a sequence of mass/charge (*m*/*z*) ratios and their intensities. The substance qualitative analysis information, such as relative molecular weight determination, chemical formula determination, and structural identification, can be obtained by comparing the measured mass spectra with the standard mass spectra manually. The peak in the mass spectra represents the distribution of ions in samples. According to the resolution difference, the mass spectral data can be divided into two types: high resolution and low resolution [2]. Low resolution mass spectra can only distinguish different nominal mass ions. High resolution mass spectra can calculate the precise mass for each ionized compound. Even isotopes can be distinguished by the high-resolution mass spectrometry. The presence of isotopes, as well as the sample purity, electronic noise, or the accuracy of mass spectrometers, will significantly affect the high-resolution mass spectral data [3]. Even under such delicately experimental conditions controlling, it is hard to obtain the identical mass spectra.

However, in real life scenarios, different compounds are often mixed together. Because of the complication of mass spectra the mathematical methods were often utilized to detect the specified compounds in the mixing samples [4]. Machine learning has been applied as an efficient tool in the analytical chemistry for a long time [5]. Partial least squares (PLS) is one of the methods used for compound detection [6]. However, when data is large-scale, PLS does not work well. Over the past several years, many works on artificial neural network (ANN) in the area of mass spectrometry have been reported [7]. MSnet, created by Curry et al. might be the earliest general neural network used in mass spectra analysis [8]. It involves a hierarchical system of several neural networks for mass spectrometry. Werther et al. compared the ANN performances using classical multidimensional numerical analysis techniques [9]. A. Eghbaldar et al. presented a methodology for the optimization of ANN to identify the compounds’ structural features from mass spectral data [10]. Ion mobility spectra could also be successfully classified by neural networks from the combination of drift time, number, intensity, and shape of peaks [11].

In general, good performance of ANN is often based on the large-scale data sets. Furthermore, the large dimension of the mass spectra input data is a natural characteristic for the ‘‘data–response’’ correlation problems [12]. Unfortunately, a small number of samples and the large dimension of inputs construct the typical dilemma for the real data sets. In analytical chemistry, principal component analysis (PCA), as a traditional method, is regularly applied for the dimension-reduction of data [13]. Although PCA has a wide range of application areas such as data compression to eliminate redundancy and data noise cancellation. PCA can only obtain principal components in a single direction. The principal components with small contribution rates may often contain important information for sample dissimilarities. In some cases, these principal components cannot be ignored. Because of the complexity and the size of the mass spectra data, automated feature extraction is frequently required to process the data [14]. Continuous wavelet transform (CWT) shows a better best performance among several peak detection algorithms for the mass spectra data [15]. Deep learning, by neural unit learning data characteristics, can extract features from raw large dimension data directly [16]. Deep learning was explored to predict molecular substructure in the mass spectral data [17]. Because the deep neural network (DNN) is not suitable for the time series data, the long short-term memory (LSTM)-based substance classification system was proposed for substance detections [18]. Nevertheless, it is still difficult to detect a class with similar molecular structure inside the mixing sample mass spectra. Felicity Allen et al. built a web server for spectrum prediction and metabolite identification from tandem mass spectra based on deep learning [19]. The multi-label classification can study each example associated with a set of labels simultaneously [20]. In this paper, we analyzed the performance of multi-label classification techniques in the detection of the small molecule metabolites found in the human body within the samples. The pragmatic algorithm was based on convolutional neural networks (CNN). Boosting is a common machine learning method, which is widely used and effective. In classification problems, it improves the performance of classification by changing the weights of training samples, learning multiple classifiers, and linearly combining these classifiers. AdaBoost, GBDT, and XGBoost are the most classic algorithms of the Boosting series. We chose XGBoost for comparison, which performed best in this work among the three algorithms. Compared with other algorithms, CNN can separate independent mass spectra contributed by different compounds to achieve better accuracy. Imitating the CNN architecture, three-channel architecture [21] has been used as input. Comparison with the traditional classification methods, the methodology can significantly enhance the predicting accuracy and performance for the mixing sample. Meanwhile, this model has a greater potential for large-target mass spectral data analysis.

## 2. Results

We have used the loss function to estimate the degree of inconsistency between the predicted value of the model and the true value. The smaller the loss function, the better is the robustness of the model. In the classification problem of machine learning, sigmoid and softmax are activation functions often used in the output layer of the neural network. They are functions used for binary and multi-class classification, but sigmoid can also be used for multi-class. The difference is that multiple classes in sigmoid may mutually overlap, softmax must be based on the premise of various types of mutual exclusion, and the sum of the probability of each category is one. Binary cross-entropy and categorical cross-entropy are the loss functions corresponding to sigmoid and softmax. In this work, sigmoid was employed as the activation function and the binary cross-entropy was employed as the loss function for multi-labels classification, which trains model fast and requires less memory:(1)$$BCE\left(x\right)=-\frac{1}{m}\overset{m}{\underset{i=1}{\sum }}y_{i}\log f_{i}\left(x\right)+\left(1-y_{i}\right)\log\left(1-f_{i}\left(x\right)\right)$$
where *x* is the input sample, *m* is the total number of training data, *y* is the real label corresponding to the *i*th category, and *f* is the corresponding model output value.

Since the output of each label is assumed to be independent, the common configuration of multi-labels binary classification is BCE and sigmoid activation function. Each category output is a probability in the range of zero to one and corresponds to a sigmoid. Adam algorithm is used as the optimizer to iteratively update neural network weights based on the training data. The main advantage of Adam is that after the offset correction, each iteration learning rate has a certain range, which makes the parameters relatively stable. After 100 epochs of training, the accuracy, recall, and precision on the test set of each model for target compounds detection are shown in Table 1.

As a result, all five machine learning models could achieve high accuracy for detecting multiple target compounds in the overlapping samples. It was indicated that they could learn features effectively from raw MS data directly. The results obtained by neural networks are often not labels such as zero or one, but they can obtain classification results such as 0.5, 0, 8. Therefore we should choose a threshold. When the result is more than the threshold, the predicted value is judged as one; when the result is less than or equal to threshold, the predicted value is judged as zero. Increasing the threshold, we will be more confident in the predicted value, which increases the precision. But this will reduce the recall. If the threshold is reduced, the number of true examples missed by the model will decrease, and the recall will increase. Here we chose 0.5 as a threshold. No matter how strong the noise was, CNN always achieved higher accuracy than the other two models. In fact, when average intensity of noise was 4 and variance is 0.8, most information of low intensity had been covered by noise. Because of good adaptive and outlier processing, CNN has the best performance of extracting features. Furthermore, for target compounds detection, the recall is the more important indicator. The recall reflects the correctly predicted components (true positive) accounts for the proportion of all components that should be predicted (true positive and true negative). Although it seems that the precision of PCA+SVM is not too low, the performance of PCA + SVM is much lower than neural networks. Because the PCA + SVM model predicts a large number of samples with positive labels as false (it does not detect the presence of compounds), it has much poorer recall performance than DNN and CNN. The precision of LSTM is high but the recall is low, it means the threshold should be reduced. XGBoost also seems to be a passable method. For further observation, we use the receiver operating characteristic (ROC) curve to evaluate the discriminative ability of the models [22]. The comparison of the ROC curves (MS data with noise that average intensity is 1 and variance is 0.2) for five models is shown in Figure 1.

Class 1: 1-methylhistidine; class 2: 1,3-diaminopropane; class 3: 2-ketobutyric acid; class 4: 2-hydroxybutyric acid; class 5: 2-methoxyestrone class 6: (*R*)-3-hydroxybutyric acid; class 7: deoxyuridine; class 8: deoxycytidine; class 9: cortexolone; class 10: deoxycorticosterone; class 11: 4-pyridoxic acid; class 12: alpha-ketoisovaleric acid; class 13: *p*-hydroxyphenylacetic acid; class 14: iodotyrosine; class 15: 3-methoxytyramine; class 16:(*S*)-3-hydroxyisobutyric acid; class 17: 3-*O*-sulfogalactosylceramide; class 18: ureidopropionic acid; class 19: tetrahydrobiopterin; class 20: biotin.

If the ROC curve was farther from the pure opportunity line (black dotted line), the model discrimination was stronger. Because of the mapping principle of the ROC curve, neural network methods and other methods have different curve shapes. According to Figure 1, CNN has the largest average area under curve (AUC), and this means CNN has the best performance of classification. PCA + SVM has the least average AUC. At the same time, we can see that all classes have large AUC using CNN, but some classes have small AUC when using DNN, LSTM, and XGBoost. Hence, it can be concluded that CNN is the robust method in the five algorithms. The average precision (AP) score can also summarize the precision-recall curves as the weighted mean of precisions achieved at each threshold to estimate five models:(2)$$\mathrm{AP}=\underset{n}{\sum }(R_{n}-R_{n-1})P_{n}$$
where $P_{n}$ and $R_{n}$ are the precision and recall at the *n*th threshold. The AP can be considered as the fraction of positive samples. In multi-label classification, mean average precision (mAP) is a common evaluation measure:(3)$$m\mathrm{AP}=\sum_{i=1}^{n}AP_{i}$$
where $AP_{n}$ is the average precision of the *n*th label. Here mAP is equal to the area under average ROC curve. Table 2 shows that PCA + SVM had poor classifier performance. Combining with Table 1 and ROC curves in Figure 1, CNN model has more stable performance of target detection for all compounds compared with DNN.

With interference data of 20 other compounds adding to mix with above simulate data (noise has 1 average intensity and 0.2 variance), the types and concentration of compounds would be random in the mixing samples.

Usually, metabolomic identification tasks may involve up to and over 100 metabolites. According to Table 3 and Table 4, when more metabolites are added, the accuracy reduction of CNN is smaller than other models. This shows that CNN is more feasible for large sample detection. Not only the 20 metabolites used in this work can be identified, more metabolites even other types of compounds can be classified using CNN.

## 3. Discussion

If more preprocesses such as smooth, baseline correction, and peak picking are applied to the data, the traditional machine learning algorithms such as SVM will perform better [15]. Compared with traditional machine learning algorithms, deep learning does not need too much preprocessing or denoising. Good performance occurs in the single mass spectral data classification using SVM or deep learning [23]. We can see the performance of the five models in Table 5:

We used 600 MS data for the test. TP means all the compounds were detected; FP means part of the compounds were detected; TN means nonexistent compounds were predicted. In multi-label target detection of mixture mass spectral data, CNN performs better than SVM XGBoost, DNN, or LSTM. The result of test MS data demonstrates that CNN is the most promising model for multi-labels target detection of mixture MS data. In fact, SVM has a good effect on two-label classification problems. It does not work well in multi-label classification problems. Compared to DNN, CNN has better performance on mixture MS data. LSTM is more suitable for time series data analysis. In the case of large noise, the classification problem may appear to overfit using Boosting methods.

Three typical MS data were chosen to analyze the CNN model prediction. In Figure 2, Figure 3 and Figure 4, on the left are the mass spectra of the mixture samples; on the right are the mass spectra of existing compounds in the mixture and the mass spectra of the predicted compounds. As shown in Figure 2, MS data consists of compound 3, compound 13, compound 18 and all the compounds that have been detected. In Figure 3, MS data consists of compound 4, compound 18, compound 15, and compound 16, but compound 4 missed in the prediction of the model, because the intensity of compound 4 is much weaker than the other three compounds. Compound 4 in this MS data is hard to be detected. In Figure 4, MS data consists of compound 6, compound 8, compound 16, and compound 18. From observation, the intensity of compound 6 and compound 16 are smaller compared to compound 8. This caused the model to predict the presence of compound 6 erroneously.

As Skarysz et al. speculated, the increased performance of CNN might be due to the following reasons. 1. The convolution filter can allow the CNN to learn the peak shape, not just the same *m*/*z* axis. 2. Such filters allow for greater robustness before low signal to noise ratios. 3. The depth of CNN may allow for a higher-level representation of the low-level features characterizing each individual pattern [23].

CNN is mainly used to identify two-dimensional graphics that are invariant to displacement, scaling, and other forms of distortion. The shape of each compound in mass spectra is similar to feature in graphics. Since the CNN feature detection layer learns from training data, explicit feature extraction is avoided, and learning is implicitly performed from the training data when using CNN. CNN has good ability to learn low-level features from complicated input. Meanwhile, CNN is less affected by noise because of the robustness of filters. If there is more MS data of different energy as input, CNN will learn more relationships between different energy. Once more MS data of different energy as input are added, even continuous signal in energy axis, further studies needed to be made to the architecture of CNN including depth, alternating layers, and filter sizes to improve the ability to learn and detect the target compounds. As an easier controllable variable factor, the energy was selected as one input channel instead of time.

The analysis of spectral data sets with small number samples is a bottleneck for deep learning including DNN, CNN, etc. Normally, those models require large data sets to learn the features of samples [12]. For this reason, deep learning may not achieve good performance in target detection when spectral data sets are small. Based on the assumption of linear mixture model, adding simulation data into data set is a method to solve the problem of the training data shortage. When dealing with MS data with different noise in Table 1, the mean average precision scores of SVM + PCA model are 0.72, 0.71, 0.68, and 0.62; the mAP scores of DNN are 0.96, 0.95, 0.92, and 0.88; the mAP scores of LSTM are 0.94, 0.92, 0.87, and 0.84; the mAP scores of XGBoost are 0.94, 0.93, 0.91, and 0.87. Meanwhile, the mAP scores of CNN are 0.99, 0.98, 0.95, and 0.92. When interference data were added, the mAP scores for the five models are 0.65, 0.80, 0.82, 0.84, and 0.95.

Compared with traditional algorithms, deep learning is more transferable and less affected by data. In real situations, owing to the differences in operator, instruments, and experimental environments, MS data may vary momentously. It is essential to control variable factor and obtain a large-scale data set for training models by data obtained under the same conditions and data expansion. Training model by input MS data from different instruments, adding more offset MS data will make models have stronger universality. Our established methodology could efficiently achieve the multiple compounds recognition from the tandem mass spectral data.

## 4. Material and Methods

### 4.1. Principal Component Analysis

PCA is a dimensionality reduction procedure. Through orthogonal transformations, PCA can convert a set of possible correlated variables into a set of values of linearly uncorrelated variables called principal components [24]. PCA expresses the data in a way as to highlight their similarities and differences. It maps n-dimensional features to k-dimensional (k < n) in a completely new orthogonal feature. Finding the k-dimensional principal component of the sample x is actually the kth eigenvalues corresponding to the eigenvector matrix P of the covariance matrix $\frac{1}{m}XX^{T}$ of the sample set x. Then for each sample x, make transformation y = Px. The purpose of PCA is to reduce the converted matrix dimension. Therefore, the number of principal components should be less than or equal to the number of original variables.

### 4.2. Support Vector Machine

As a non-parametric technique, support vector machine (SVM) constructs an optimal hyperplane decision function in feature space that is mapped from the original input space. Support vectors are data points that are closer to the hyperplane and influent the position and orientation of the hyperplane. Support vectors will decide the maximum margin related to hyperplane. Jian Cui et al. proposed a matching algorithm with isotope distribution pattern in LC-MS based on SVM learning model [25]. However, when the data set is large, the kernel function mapping dimension will be high. With the calculation amount increasing, the input data often need dimensionality reduction before SVM. Ressom et al. used ant colony optimization for peak selection and SVM for classification [26]. Yang et al. used recursive feature elimination for peak selection [15]. In this paper, SVM with PCA was chosen for the comparison. Sunil Kr. Jha et al. used SVM with PCA for human body odor discrimination and achieved 0.86 maximum class recognition [27].

### 4.3. Artificial Neural Network

ANN is a framework for many different machine learning algorithms to work together and process complex data inputs. In the ANN, a simple model will be established to form different networks according to different connection methods. An artificial neural network is an operational model consisting of a large number of neurons connected to each other, as shown in Figure 5. Each neuron represents a specific output function called activation function. The connection between every two nodes represents a weighting value for passing the connection signal, called weight, which is equivalent to the memory of the artificial neural network. The output of the network varies depending on the connection method of the network, the weight value, and the activation function. The network is usually an approximation of an algorithm or function in nature or a logic strategy expression [28].

### 4.4. Deep Neural Network

Deep neural network (DNN) is developed from ANN. The “deep” in “deep neural network” refers to the greater number of layers through which the data is transformed [16]. Generally, the number of hidden layers is more than five. In DNN, each layer of nodes trains on a distinct set of features based on the previous layer’s output. Each step of the neural network involves estimator. It makes error measurement to form a slight update of its weight. With an incremental adjustment, the coefficient factors can slowly learn to pay attention to the most important features. The back-propagation algorithm is widely used to update weight with a gradient descent approach.

### 4.5. Convolutional Neural Network

Based on the architecture of DNN, convolutional neural network (CNN) consists of three structures: convolution, activation, and pooling. For saving spatial information about the data, the sub-matrices of adjacent input neurons are concatenated into single hidden layer neurons belonging to the next layer. The single hidden layer of neurons represents a local receptive field. This operation is called “convolution” as shown in Figure 6. With the sharing of convolution kernel, CNN has no pressure on high-dimensional data processing. Meanwhile, CNN can obtain excellent feature classification effects without manually selecting features. CNN can also exploit the data geometrical properties without getting affected by the noise compared to other techniques [29].

### 4.6. Mass Spectrum Data

Assuming the mixture mass spectra are equal to the weighted sum of the mass spectra of the individual compounds [30,31]. The mass spectra of mixtures may be represented by the array:(4)$$x_{i,j}=\overset{n}{\underset{k=1}{\sum }}a_{ik}s_{kj}$$
where $x_{i,j}$ is the intensity of the *i*th mass in the *j*th mixture, *n* is the number of components in the mixture, $a_{ik}$ is the intensity of mass *i* in pure compound *k*, and $s_{kj}$ is the concentration of compound *k* in the source for mixture *j*. Equation (4) can be simply expressed as:(5)$$X_{m}=AS$$
where $X_{m}$ is an i×j matrix, A is an i×k matrix, and S is an k×j matrix; i, j, k respectively represent the number of different mixtures, masses, and pure compounds and X, A, S respectively represent overlapping mass spectral data matrix, pure compound matrix, and concentration matrix.

According to the assumption of the linear mixture model, the peaks from several spectra can be combined into a single spectrum. If the difference between their m/z values is less than 10 ppm, the intensity and the m/z value from the input spectrum will be aggregated into a single peak. Meanwhile, to present chemical noise, Gaussian noise is partially added to simulate the real data [32]:(6)$$X=X_{m}+X_{n}$$
where X is the simulation data matrix used to train and test model, $X_{m}$ is overlapping mass spectral data matrix, and $X_{n}$ is Gaussian noise matrix. Different instruments, environments, and experimental operations often produce different noises. It is not possible to fully simulate real noise data, but the signal-to-noise ratio of simulate data is close to or greater than real data. In order to test the denoising ability of the models, different intensities of noise is chosen to be added into the models, as shown in Figure 7.

In order to simulate mass spectral data of small molecule metabolites found in the human body, 20 pure compounds of LC-MS/MS positive data (1-methylhistidine, 1,3-diaminopropane, 2-ketobutyric acid, etc.) from Human Metabolome Database (HMDB) were chosen as target labels with three energies (10, 20, 40 eV) [33]. HMDB is a web-enabled metabolomic database containing comprehensive information about human metabolites. Compared to LC-MS data, the tandem mass spectra (LC-MS/MS) include more molecular structure information that would be applied to reduce the impurity interferences in the data. Based on a linear mixture model, 3000 simulation data with stochastic concentration (the range 0 to 100%) of compounds were generated. The data set was divided into 1920 training sets, 480 validation sets, and 600 test sets. In label data, 1/0 was set to indicate the presence/absence of the specified compound. Figure 8 and Figure 9 respectively show experimental and simulate LC-MS/MS spectra, and they have great similarities. Therefore, simulate data can largely represent real data.

### 4.7. Methods

According to previous studies, the effects of extraneous information were generally minimized by data pre-processing steps (smoothing, baseline correction, normalization, peak selection, and peak calibration) [9]. Feature selection or dimensionality reduction is significant for traditional machine learning algorithms. With more features, those traditional algorithms may spend more time in analyzing features and training the model. It causes “dimensional disasters” to engender a much complex model. In order to collect the self-information rather than the extraneous variables and noise such as stray light in the background of the sample, an appropriate algorithm for feature selection is essential. It can conduct a good correlation between the spectral information and the content value. PCA is also commonly used with SVM for analyzing MS data. As an end-to-end algorithm, DNN can directly deal with raw data, denoising and extracting features synthetically. In this paper, the particular CNN was exploited to learn to recognize overlapping compounds directly from raw MS data. However, DNN is an algorithm based on large-scale data sets. It is difficult to obtain sufficient MS data in the actual scene. The simulation data was selected to train the model. Finally, the trained models were applied to identify target compounds in the test samples. The Keras and Scikit-learn python modules were also employed to build models [34,35]. The architecture of DNN model is shown in Table 6.

The MS data of three energies were flatten as one-dimensional input. Based on DNN, the established CNN model had same number of layers and same fully connected layers with rectified linear units (ReLU) activation compared with DNN. The differences between CNN and DNN were the usage of convolution layers and the dimension of input. The architecture of CNN model is shown in Table 7.

Utilizing the 900 *m*/*z* windows with 10 ppm interval, in the range from 17 to 875 *m*/*z*, the MS data of each energy could be flattened into a one-dimensional matrix. With the character of input, one-dimensional convolution layers could further simplify the model. The information between different energy could be recognized by the convolution layers. Max pooling layers were applied to abstract the characteristics of the region and reduce the coupling degree of the model. Convolution layers and pooling layers were used for feature extraction. The fully-connected layers were applied for classification. In the last fully-connected layer, sigmoid activation was applied to output the probability of each compound presence. Usually the model threshold was set to 0.5. The threshold value also could be modified according to the actual condition. The operation flow is shown in Figure 10.

## 5. Conclusions

In this paper, we proposed a methodology based on CNN to detect multiple compounds directly from the LC-MS/MS mixture sample data. By evaluating the different neural network architectures, the mean average precision of CNN could achieve 95% to 97%. Compared with DNN and traditional PCA with SVM method, CNN has the best performance on accuracy, recall, and precision. However, the current CNN model can only detect the existing compounds. The concentration prediction will be considered in further research. More circumstances also need to be reflected, such as spectral skewing which will influence the quantitative analysis. Benefiting from the good portability and the generalization of neural networks, models can be transplanted into mobile terminals for real-time detection of soil pollution, disease detection, and food safety, etc. This method has broad prospects in real life multiplex compound detection scenarios.

## Figures and Tables

**Figure 1 molecules-24-04590-f001:**
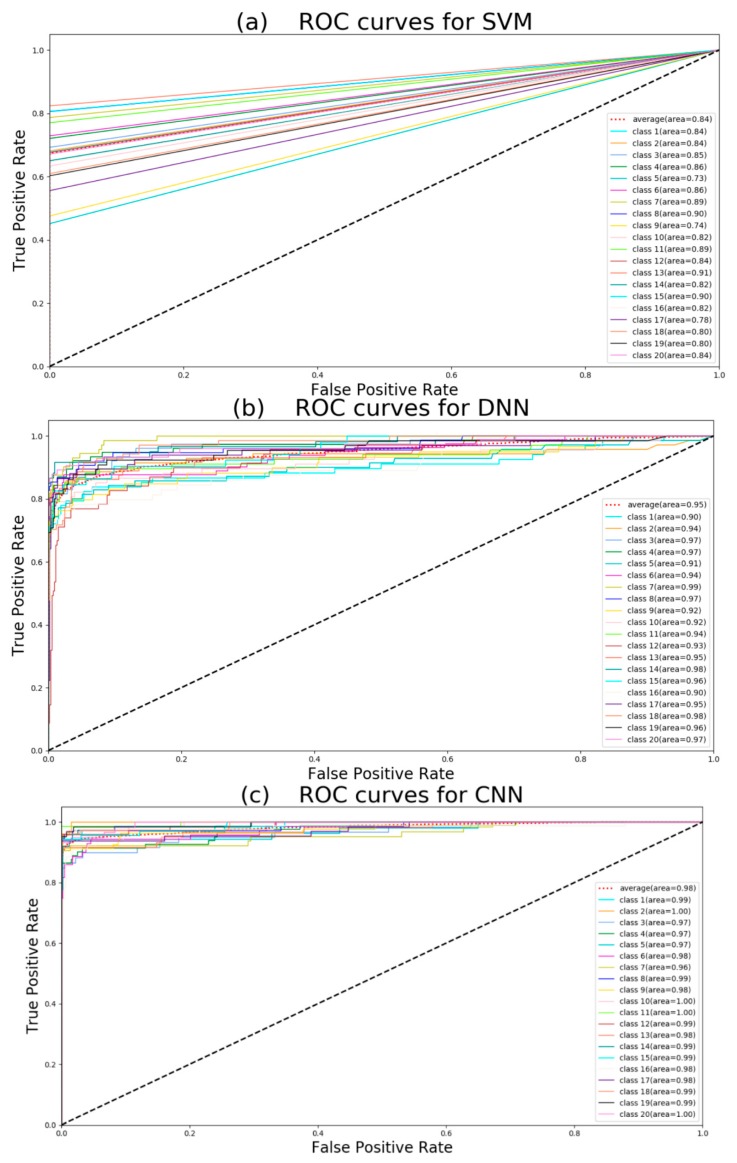
Receiver operating characteristic (ROC) curves for (**a**) principal component analysis (PCA) + support vector machine (SVM); (**b**) deep neural network (DNN); (**c**) convolutional neural network (CNN); (**d**) long short-term memory (LSTM); (**e**) XGBoost.

**Figure 2 molecules-24-04590-f002:**
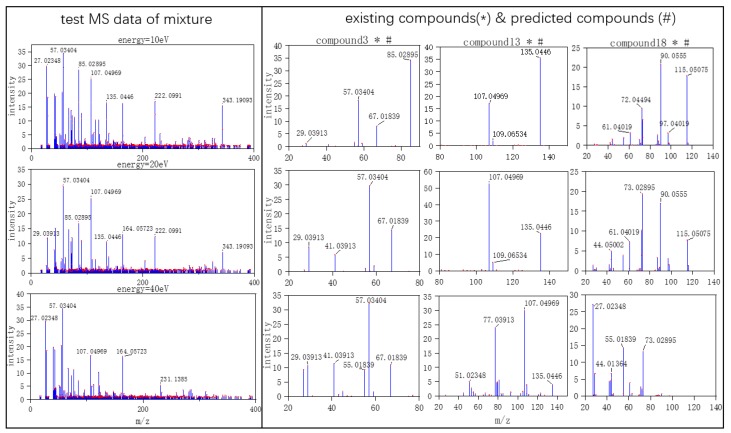
True positive mass spectrometry (MS) data (the selected MS data were labeled by red dots).

**Figure 3 molecules-24-04590-f003:**
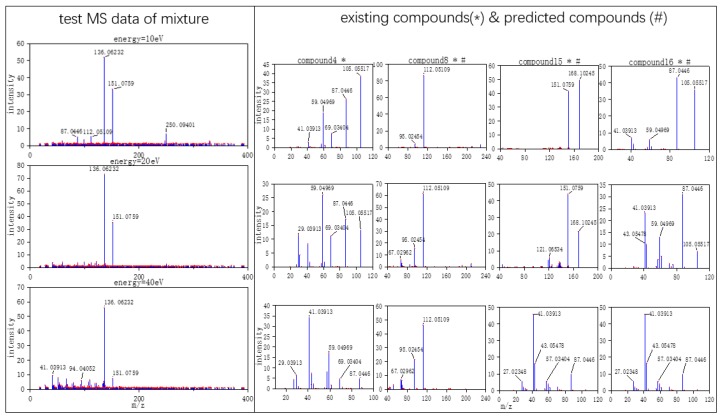
False positive MS data (the selected MS data were labeled by red dots).

**Figure 4 molecules-24-04590-f004:**
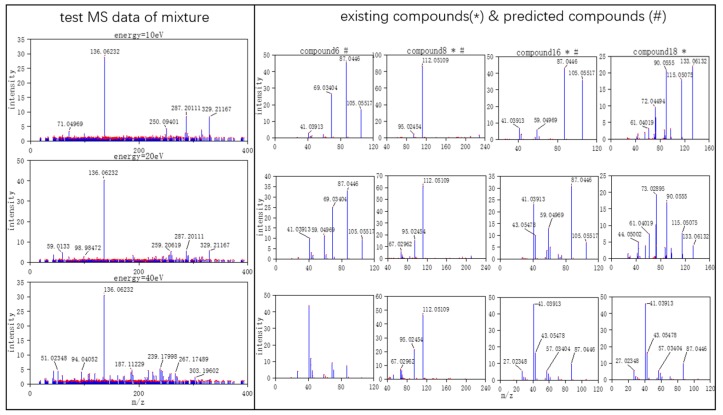
True negative MS data (the selected MS data were labeled by red dots).

**Figure 5 molecules-24-04590-f005:**
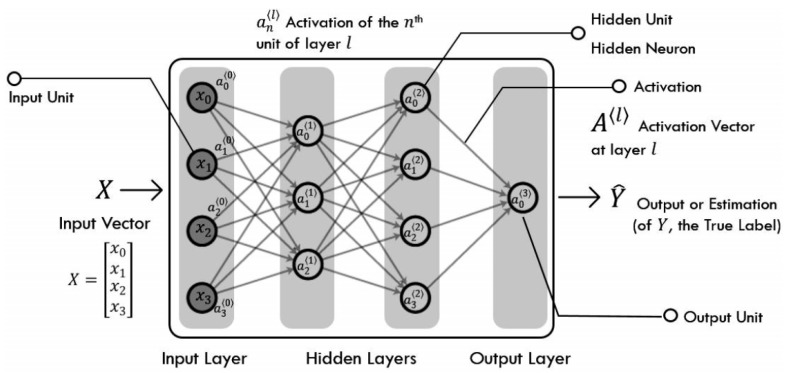
Artificial neural network: The artificial neural network chooses the weight of the network reasonably according to the specific loss information.

**Figure 6 molecules-24-04590-f006:**
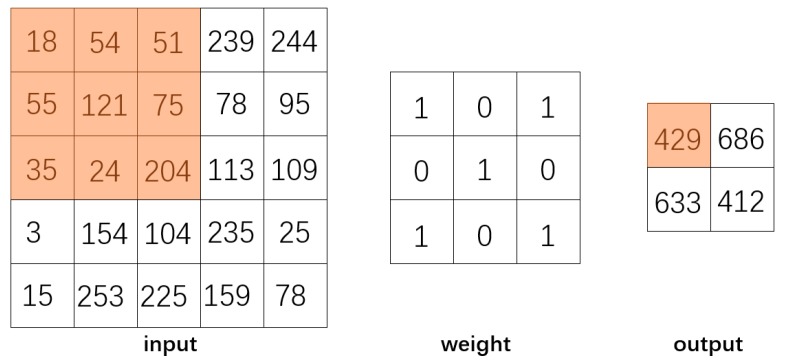
Convolution operation: By sliding the convolution kernel over the input and computing the dot product then get a matrix called the convolution feature.

**Figure 7 molecules-24-04590-f007:**
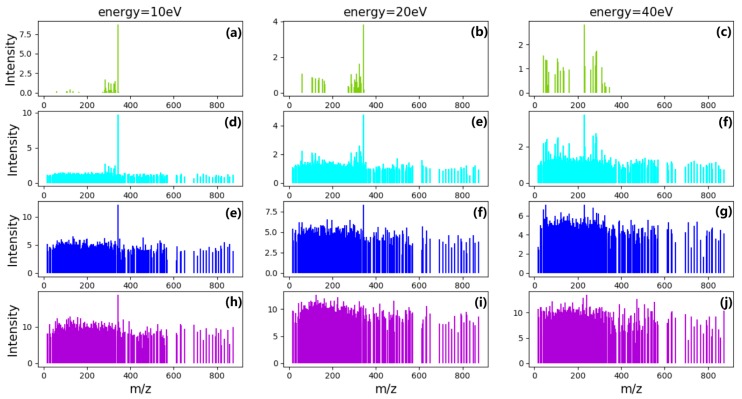
Mass spectra of different energy before and after adding noise. (**a**–**c**) are MS data without noise; (**d**–**f**) are MS data with noise that average intensity is 1 and variance is 0.2; (**g**–**i**) are MS data with noise that average intensity is 4 and variance is 0.8; (**j**–**l**) are MS data with noise that average intensity is 8 and variance is 1.6.

**Figure 8 molecules-24-04590-f008:**
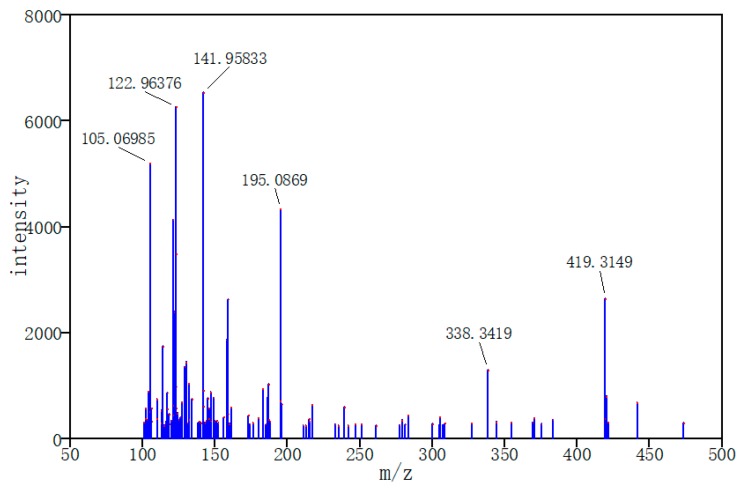
Experimental tandem mass spectra (LC-MS/MS)-ESI-QTOF (the selected MS data were labeled by red dots).

**Figure 9 molecules-24-04590-f009:**
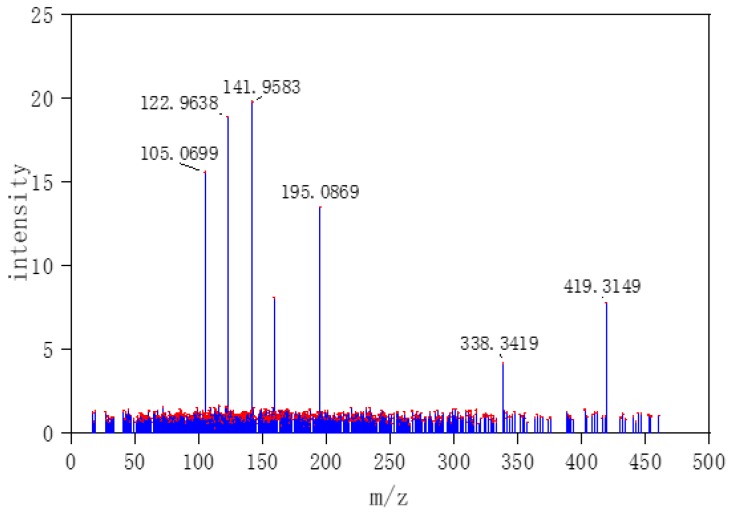
Simulate LC-MS/MS spectra (the selected MS data were labeled by red dots).

**Figure 10 molecules-24-04590-f010:**
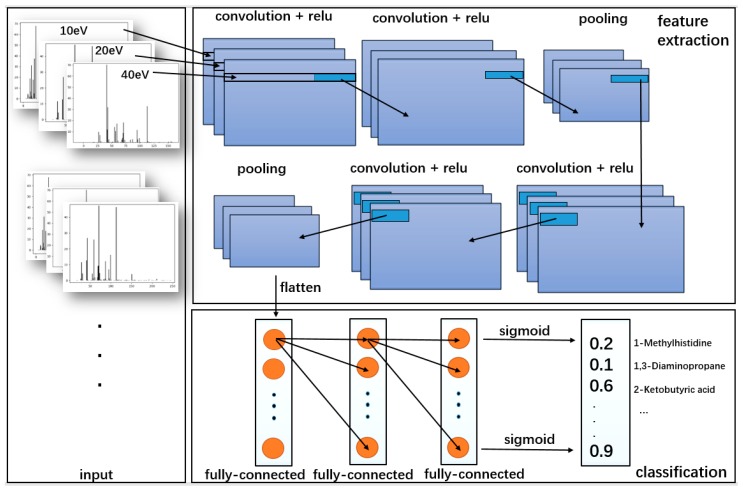
The flow of CNN: The convolutional neural network adopts MS data as input, which can effectively learn corresponding features from a large number of samples and avoid complex feature extraction process.

**Table 1 molecules-24-04590-t001:** Accuracy(A)/recall(R)/precision(P) for the five models.

Number	PCA + SVM			DNN			CNN			LSTM			XGBoost		
	A	R	P	A	R	P	A	R	P	A	R	P	A	R	P
1^a^	0.93	0.55	0.73	0.97	0.83	0.61	0.99	0.89	0.67	0.95	0.39	0.87	0.98	0.72	0.64
2^b^	0.93	0.56	0.74	0.95	0.86	0.51	0.97	0.87	0.64	0.94	0.37	0.84	0.97	0.70	0.62
3^c^	0.92	0.54	0.74	0.94	0.88	0.48	0.95	0.87	0.62	0.91	0.36	0.82	0.94	0.67	0.60
4^d^	0.90	0.50	0.75	0.92	0.84	0.49	0.94	0.86	0.60	0.90	0.34	0.78	0.88	0.62	0.58

1^a^: MS data without noise; 2^b^: MS data with noise (average intensity = 1, variance = 0.2); 3^c^: MS data with noise (average intensity = 4, variance = 0.8); 4^d^: MS data with noise (average intensity = 8, variance = 1.6).

**Table 2 molecules-24-04590-t002:** Mean average precision score for the five models.

Models	PCA + SVM	DNN	CNN	LSTM	XGBoost
mAP	0.84	0.95	0.98	0.94	0.94

**Table 3 molecules-24-04590-t003:** Accuracy(A)/recall(R)/precision(P) for the five models (added interference data).

PCA + SVM			DNN			CNN			LSTM			XGBoost		
A	R	P	A	R	A	R	P	A	R	A	R	P	A	R
0.93	0.54	0.71	0.94	0.91	0.93	0.54	0.71	0.94	0.91	0.93	0.54	0.71	0.94	0.91

**Table 4 molecules-24-04590-t004:** Mean average precision score for the five models (added interference data).

Models	PCA + SVM	DNN	CNN	LSTM	XGBoost
mAP	0.65	0.80	0.95	0.82	0.84

**Table 5 molecules-24-04590-t005:** Number of true positive (TP)/false positive (FP)/true negative (TN) test data for the five models.

Number	Models					
	PCA+SVM	DNN	CNN	LSTM	XGBoost	PCA+SVM
TP	282	293	451	270	402	TP
FP	318	284	142	198	168	FP
TN	0	23	7	132	30	TN

**Table 6 molecules-24-04590-t006:** The architecture of DNN model (N: the number of *m*/*z* windows).

Layer (Type)	Output Shape	Parameters
dense_1 (Full-connected)	3 * N	7325142
batch_normalization_1	3 * N	10824
dense_2 (Full-connected)	3 * N	7325142
batch_normalization_2	3 * N	10824
dense_3 (Full-connected)	1.5 * N	3662571
batch_normalization_3	1.5 * N	5412
dense_4 (Full-connected)	1.5 * N	1831962
batch_normalization_4	1.5 * N	5412
dense_5 (Full-connected)	0.6 * N	732514
batch_normalization_5	0.6 * N	2164
dense_6 (Full-connected)	0.3 * N	146340
batch_normalization_6	0.3 * N	1080
dense_7 (output)	20	5420

**Table 7 molecules-24-04590-t007:** The architecture of CNN model (N: the number of *m*/*z* windows).

Layer (Type)	Output Shape	Parameters
conv1d_1 (Conv)	N, 32, 3	512
conv1d_2 (Conv)	N, 32, 3	5152
max_pooling1d_1(Pooling)	N//2, 32, 3	0
conv1d_3 (Conv)	N//2, 64, 3	6208
conv1d_4 (Conv)	N//2, 64, 3	12352
max_pooling1d_2(Pooling)	N//4, 64, 3	0
flatten_1 (Flatten)	16 * N	0
dense_1 (Full-connected)	N//5	7721693
dense_2 (Full-connected)	N//10	146340
dense_3 (Full-connected)	20	5420

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
