# Peer review of "Multiple Compounds Recognition from The Tandem Mass Spectral Data Using Convolutional Neural Network"

_molecules, 2019, doi:10.3390/molecules24244590_

Round 1
Reviewer 1 Report
This work presents the use of convolutional neural networks to predict the amounts of 20 different metabolites in a synthetic dataset. The authors represent spectral intesnsity data as an N x 3 array, where N is the number of mass bins, and 3 is the 10eV, 20eV, 40eV modes respectively. The scope of the model is limited to molecules for which the MS/MS spectra already exists.
I have a couple of critiques:
I am not sure how representative the synthetic data are to real metabolomic identification tasks using MS/MS. Metabolomic identification tasks may involve up to and over 100 metabolites. Does the classification scale beyond the 20 metabolites used in the data set here? The noise generated for the training set was all Gaussian noise. How certain are you that the noise for real spectra is also Gaussian? It would have been helpful if the authors had included a real experimental example. Make samples using different combinations of the 20 metabolites, take an MS/MS reading of these molecules, and see how well it performs. How does the convolutional network structure help with identification of the spectra? Are the gains of the CNN model mostly due to the reduced number of parameters? What would have happened if you used an LSTM architecture instead? I don’t really understand Figures 2-4: Are these meant to show single examples of correctly predicted spectra, false negatives in spectra, and false positives respectively? It would be helpful to list the correct amounts of each compound.
The following work also uses a machine learning model for peak identification and compound analysis in MS/MS spectra, and should also be cited:
Felicity Allen, Allison Pon, Michael Wilson, Russ Greiner, David Wishart, CFM-ID: a web server for annotation, spectrum prediction and metabolite identification from tandem mass spectra, Nucleic Acids Research, Volume 42, Issue W1, 1 July 2014, Pages W94–W99, https://doi.org/10.1093/nar/gku436
Here are a few typos I noticed:
20: false -> falsely
21: miss acknowledged -> misidentified
77-78: incomplete sentence
108: loud -> strong
152 - 153: ‘Accord … less likely to increase for CNN.’ -> awkward sentence
342: existed -> existing
Author Response
Dear Editor and Reviewers,
Thank you for your useful comments and suggestions for our manuscript titled “Multiple Compounds Recognition from The Tandem Mass Spectral Data Using Convolutional Neural Network”. We have revised the manuscript accordingly. Detailed responses to the points raised by the referees are listed below point by point:
This work presents the use of convolutional neural networks to predict the amounts of 20 different metabolites in a synthetic dataset. The authors represent spectral intesnsity data as an N x 3 array, where N is the number of mass bins, and 3 is the 10eV, 20eV, 40eV modes respectively. The scope of the model is limited to molecules for which the MS/MS spectra already exists.
I have a couple of critiques:
I am not sure how representative the synthetic data are to real metabolomic identification tasks using MS/MS. Metabolomic identification tasks may involve up to and over 100 metabolites. Does the classification scale beyond the 20 metabolites used in the data set here?A: We also performed classification experiments based on other different metabolites and found that they also have good accuracy. In fact, we can arbitrarily choose a reasonable number of different metabolites for identification. The premise is that the amount of data obtained is sufficient and the proportion of different metabolites is reasonable. But this is often difficult. Therefore, it is often necessary to perform data expansion based on real data.
The noise generated for the training set was all Gaussian noise. How certain are you that the noise for real spectra is also Gaussian? It would have been helpful if the authors had included a real experimental example. Make samples using different combinations of the 20 metabolites, take an MS/MS reading of these molecules, and see how well it performs.
A: Different instruments, environments, and experimental operations often produce different noises. Therefore, it is not possible to fully simulate real noise data. This comparison experiment is to show that the trained model has good denoising ability. Gaussian noise is a relatively common noise. We can also add other noises to interfere with the classification of the model. Here we add an experimental example for contrast.
How does the convolutional network structure help with the identification of the spectra? Are the gains of the CNN model mostly due to the reduced number of parameters? What would have happened if you used an LSTM architecture instead?
A: Convolutional neural networks are mainly used to identify two-dimensional graphics that are invariant to displacement, scaling, and other forms of distortion. The shape of each compound in mass spectra is similar with feature in graphics. Since the CNN feature detection layer learns from training data, when using CNN, explicit feature extraction is avoided, and learning is implicitly performed from the training data.
CNN reduces the number of weights that need to be trained and reduces the computational complexity of the network through weight sharing. At the same time, through the pooling operation, the local transformation of the network to the input has certain invariance such as translation invariance, scaling invariance, etc., which improves the generalization ability of the network.
We also tried LSTM for classification and found it has worse performance than CNN. As a comparison, we have added it in the paper already.
I don’t really understand Figures 2-4: Are these meant to show single examples of correctly predicted spectra, false negatives in spectra, and false positives respectively? It would be helpful to list the correct amounts of each compound.
A: In Figure2-4, mass spectra on the left are mixture samples and mass spectra on the right are the predicted result (#) and the compounds which existing in mass spectra on the left(*). These are meant to show single examples of correctly predicted spectra, false negatives in spectra, and false positives respectively. We have replaced the figures and added some sentences to explain these figures.
The following work also uses a machine learning model for peak identification and compound analysis in MS/MS spectra, and should also be cited: Felicity Allen, Allison Pon, Michael Wilson, Russ Greiner, David Wishart, CFM-ID: a web server for annotation, spectrum prediction and metabolite identification from tandem mass spectra, Nucleic Acids Research, Volume 42, Issue W1, 1 July 2014, Pages W94–W99, https://doi.org/10.1093/nar/gku436
A: This paper has been added in article for citation.
Here are a few typos I noticed:
20: false -> falsely
21: miss acknowledged -> misidentified
77-78: incomplete sentence
108: loud -> strong
152 - 153: ‘Accord … less likely to increase for CNN.’ -> awkward sentence
342: existed -> existing
A: We tried our best to correct some grammatical errors.
Reviewer 2 Report
In this paper the author proposed a novel Multiple Compounds Recognition strategy, Multiple Compounds Recognition from The Tandem Mass Spectral Data Using Convolutional Neural Network, which has a better performance compared existing methods. In this newly computational framework, the authors have taken the Tandem Mass Spectral Data into the consideration. This is a new improvement of deep learning algorithm scheme, using in a new application domain, and the experimental results show that the proposed algorithm gains a better performance.
In general, the idea of this paper is relatively new and shows application value, the experiment designs reasonably, but the organization of this paper is not so well. And there are some detailed problems:
In the result section, the authors mentioned that they use ‘categorical cross entropy’ as their loss function. Why do they use this as their function? Could they provide some illustration or extra experiments to give a demonstration. Why are there many differences between the sub-picture (a) of picture 1 and others’ in picture 2? There might be lots of other excellent classifier such as GDBT, Adaboost and xGboost, why do the author just us the SVM+PCA as their comparison? The quality of picture Figures 2, could replace it with high quality figure. Please check the grammar and spelling mistakes carefully. Some symbols should use Italics style in this study.Author Response
Thank you for your useful comments and suggestions for our manuscript titled “Multiple Compounds Recognition from The Tandem Mass Spectral Data Using Convolutional Neural Network”. We have revised the manuscript accordingly. Detailed responses to the points raised by the referees are listed below point by point:
In this paper, the author proposed a novel Multiple Compounds Recognition strategy, Multiple Compounds Recognition from The Tandem Mass Spectral Data Using Convolutional Neural Network, which has a better performance compared to existing methods. In this newly computational framework, the authors have taken the Tandem Mass Spectral Data into consideration. This is a new improvement of deep learning algorithm scheme, using in a new application domain, and the experimental results show that the proposed algorithm gains a better performance.
In general, the idea of this paper is relatively new and shows application value, the experiment designs reasonably, but the organization of this paper is not so well. And there are some detailed problems:
In the result section, the authors mentioned that they use ‘categorical cross entropy’ as their loss function. Why do they use this as their function? Could they provide some illustration or extra experiments to give a demonstration.A: Thank you for reminding. It's a mistake using 'categorical cross-entropy’ here. Sigmoid and softmax are activation functions often used in the output layer of the neural network. Generally speaking, they are functions used for two and multi-class classification, but sigmoid can also be used for multi-class. The difference is that multiple classes in sigmoid may be mutually overlap, softmax must be based on the premise of various types of mutual exclusion, and the sum of the probability of each category is 1. One advantage of cross-entropy as a loss function is that using the sigmoid function can avoid the problem of reducing the learning rate of the mean square error loss function during gradient descent because the learning rate can be controlled by the output error. Binary cross-entropy and categorical cross-entropy are loss functions corresponding to sigmoid and softmax. Therefore, it’s better to ‘use binary cross-entropy’ instead of ‘categorical cross-entropy’ here.
Why are there many differences between the sub-picture (a) of picture 1 and others’ in picture 2?
A: For the binary classification problem (here it is a binary classification problem for the existence of a certain substance), the results obtained by some classifiers such as neural networks are often not labels such as 0 and 1, but they can obtain classification results such as 0.5, 0, 8. At this time, we artificially take a threshold, such as 0.4, then less than 0.4 is class 0, and greater than or equal to 0.4 is class 1. This threshold we can take 0.1, 0.2 and so on. Taking different thresholds, the final classification situation is different.
When making a ROC curve based on samples, you can see that the curve is a "climbing" every time. When you encounter a positive example, you climb up one grid, and when you are wrong, you climb one grid to the right. We set the threshold high at the beginning. Afterward, by continuously lowering the threshold, the samples determined as positive by the algorithm include all samples. At this time, different sets of P and R values ​​can be obtained. With the precision P as the horizontal axis and the recall R as the vertical axis, these P-R arrays are plotted on the coordinate axis, and then connected to form a curve to obtain the corresponding P-R curve diagram. Each corner point / step on the curve is a threshold.
Some classifiers such as SVM and GBDT give predictions of 0 or 1. Therefore, their ROC curve is a polyline.
There might be lots of other excellent classifier such as GDBT, Adaboost and xGboost, why do the author just us the SVM+PCA as their comparison?
A: In fact, SVM and PCA are methods proposed in other literatures usually in the domain of mass spectra, but we rarely found paper which using boosting methods for analysis of mass spectra. Boosting methods including GDBT, Adaboost and XGboost often have good performance for classification. But when noise is excessive, boosting methods may have overfitting problems. We have added XGBoost for comparison in the paper since the performance of three boosting methods are similar.
The quality of picture Figures 2, could replace it with high quality figure.A: This picture has been replaced.
Please check the grammar and spelling mistakes carefully. Some symbols should use Italics style in this study.A: We tried our best to correct some grammatical errors.
Round 2
Reviewer 2 Report
The paper is in good shape and I recommend to accept.